# Deposition of Lead Phosphate by Lead-Tolerant Bacteria Isolated from Fresh Water near an Abandoned Mine

**DOI:** 10.3390/ijms23052483

**Published:** 2022-02-24

**Authors:** Yugo Kato, Satoshi Kimura, Toshihiro Kogure, Michio Suzuki

**Affiliations:** 1Department of Applied Biological Chemistry, Graduate School of Agricultural and Life Sciences, The University of Tokyo, Tokyo 113-8657, Japan; katoh.yugo@gmail.com (Y.K.); skimura@g.ecc.u-tokyo.ac.jp (S.K.); 2Department of Earth and Planetary Science, Graduate School of Science, The University of Tokyo, Tokyo 113-0033, Japan; kogure@eps.s.u-tokyo.ac.jp; 3Collaborative Research Institute for Innovative Microbiology, The University of Tokyo, Tokyo 113-8657, Japan

**Keywords:** bacteria, bioremediation, lead, metal nanoparticles, screening

## Abstract

Specialist bacteria can synthesize nanoparticles from various metal ions in solution. Metal recovery with high efficiency can be achieved by metal-tolerant microorganisms that proliferate in a concentrated metal solution. In this study, we isolated bacteria (*Pseudomonas* sp. strain KKY-29) from a bacterial library collected from water near an abandoned mine in Komatsu City, Ishikawa Prefecture, Japan. KKY-29 was maintained in nutrient medium with lead acetate and synthesized hydrocerussite and pyromorphite nanoparticles inside the cell; KKY-29 also survived nanoparticle synthesis. Quantitative PCR analysis of genes related to phosphate metabolism showed that KKY-29 decomposed organic phosphorus to synthesize lead phosphate. KKY-29 also deposited various metal ions and synthesized metal nanoparticles when incubated in various metal salt solutions other than lead. The present study considers the development of biotechnology to recover lead as an economically valuable material.

## 1. Introduction

Elements are largely divided into metallic and nonmetallic elements. Some metallic elements, called essential elements, are necessary for maintaining metabolism and organismal homeostasis. On the other hand, both essential and non-essential elements can be toxic if ingested in excess. 

Since their discovery, humans have employed metals in all practices of life. Heavy metals have become particularly important in the modern industrialized landscape, where they have been utilized in various products such as paints, batteries, computers, and fertilizers. However, their widespread use exacerbates environmental pollution. Heavy metals have a tremendous impact on microorganisms, plants, and humans. For example, heavy metals accumulate in seeds and cause physiological dysfunctions and malnutrition in plants [1]. Heavy metals accumulate in plants and animals and cause irreversible damage at the cellular level, where they inhibit enzyme activity, disturb the functioning of essential elements, and promote the prevalence of reactive oxygen species [2,3,4].

Because lead has a low melting point and is easy to manipulate, it has seen frequent usage in the production of alloys since ancient times as well as batteries, paints, and water pipes more recently [5]. However, lead ions are toxic and can inhibit enzyme activity by coordinating with thiol groups in proteins [6,7,8,9,10]. Lead ions are more destructive in children, where they can cause neuropathy [11]. They have also been shown to have various detrimental effects on fetus development [12].

Microorganisms incorporate heavy metal elements in metabolic and homeostatic processes. For example, iron-reducing bacteria obtain energy by exchanging electrons with iron [13]; magnetotactic bacteria synthesize iron nanoparticles in the cells to detect magnetic fields [14,15]. 

Metal nanoparticles have unique physical properties characterized by the quantum size effect. They are used in medical and engineering fields as drug delivery agents, catalysts for chemical reactions, environmental cleanup agents, and antibacterial agents [16,17,18,19,20,21]. Research has also considered the value of microorganisms in synthesizing economically important metals as a novel method that requires comparatively little energy and can be incorporated in bioremediation strategies. Some bacteria with heavy metal tolerance can synthesize heavy metal nanoparticles and contribute to bioremediation [22,23,24,25]. Recovery of heavy metals by nanoparticle-synthesizing microorganisms has also been shown to mitigate heavy metal toxicity in plants [26]. However, the mechanism of nanoparticle synthesis remains largely unknown [27].

The present study investigated metal-resistant microorganisms able to synthesize metal nanoparticles, and aimed at describing the mechanisms involved. In order to isolate microorganisms with lead tolerance and lead nanoparticle synthesis abilities, we collected fresh water from around the Okoya mine in Komatsu City, Ishikawa Prefecture, and the Kamioka mine in Hida City, Gifu Prefecture, Japan. The Okoya mine closed in 1972. The river around Okoya mine is polluted with heavy metals, including copper (Cu), zinc (Zn), lead (Pb), and cadmium (Cd) [28]. Kamioka mine is also an abandoned Cu, Zn, Pb, and Cd mine. The pollution caused itai-itai disease, which is one of the four major types of pollution-derived diseases in Japan [29]. The comprehensive description of the mechanisms involved in nanoparticle synthesis could inform the development of bioremediation strategies and metal ion recovery.

## 2. Results

### 2.1. Isolation of Bacteria from an Abandoned Mine

Fresh water was collected from around the Kamioka and Okoya mines in Japan. The water samples were spread on nutrient agar medium containing lead acetate. From the medium, 96 lead-resistant microorganisms were isolated and named KKY-1 to KKY-96. The isolated strains were separate into two groups according to whether they were incubated in nutrient medium containing lead acetate (Pb(+) condition) or not (Pb(−) condition) (Figure 1). After incubation for 24 h, fluorescence intensity was measured to confirm the synthesis of lead nanoparticles. Figure 2 shows the difference in fluorescence intensity between Pb(+) and Pb(−).

To screen the strains, 11 strains with large differences in fluorescence intensity were identified by transmission electron microscopy (TEM) (Figure 3). No particles were observed in KKY-1, 6, 27, and 43. Micro particles (>100 nm in diameter) were observed at KKY-85 and 95. Nanoparticles were observed in the other five strains around the cells. To remove the effect of the growth medium in nanoparticle synthesis, the cells of KKY-29, 31, 46, 70, and 84 were incubated with lead acetate solution after washing with distilled water and observed by TEM (Figure 4). Here, nanoparticles were observed in KKY-29, 46 and 84. A large number of nanoparticles were observed in only KKY-29 under medium-free conditions. Based on these observations, KKY-29 was selected for further analysis.

To identify KKY-29, 16S rRNA was obtained by polymerase chain reaction (PCR) using universal primers, and DNA sequencing of 16S rRNA was performed. We obtained seven species with high homology from the NCBI Ribosomal Database Project (RDP), and generated a phylogenetic tree by the neighbor-joining method using MEGA software (Figure 5). The results showed that *Pseudomonas koreensis* JCM 14769 was the closest species to KKY-29, with a homology of 99.5%. 

### 2.2. Appropriate Conditions for Synthesis of Lead Nanoparticles

To explore the conditions for the synthesis of nanoparticles by KKY-29, the cells were washed with sterile water and incubated in lead acetate solution at various concentrations. The correlation between OD_600_ and dry bacterial cell weight was calculated (Appendix A). Each bacterial cell was observed by TEM (Appendix A). Nanoparticles with high electron densities were also observed. To measure the lead concentration efficiency of KKY-29, the concentration of lead in the supernatant was measured by inductively coupled plasma mass spectrometry (ICP-MS) (Appendix A). KKY-29 at 100 and 500 mg/L could not remove lead from a 5 mM lead acetate solution. It was possible to remove lead from the 0.5 and 0.05 mM lead acetate solutions. At 1000 mg/mL, KKY-29 removed more than 90% of the lead from the 0.05 mM lead acetate solution. Further experiments were conducted at bacterial concentrations of 1000 mg/L and a lead acetate concentration of 5.0 mM. Under these conditions, the average particle size was 28.0 ± 9.2 nm. The most frequently occurring size was approximately 30 nm (Appendix A). The sizes of the synthesized particles were measured using ImageJ software.

### 2.3. Identification of Mineral Species

To identify the synthesized minerals, the KKY-29 cells were observed by TEM after incubation with lead acetate. Nanoparticles were densely deposited on some of the cells, along their shape (Figure 6A). The nanoparticles were a few tens of nanometers in size and rather ellipsoidal. Energy-dispersive X-ray spectroscopy(EDS) analysis showed that they were composed of Pb, O, P, and Cl (Figure 6B). Selected-area electron diffraction (SAED) from the aggregates of the nanoparticles showed a ring pattern (Figure 6C), which could be explained well by the calculated ring pattern (top-right of the figure) for pyromorphite (Pb_5_(PO_4_)_3_Cl), using crystallographic parameters reported by Mills et al. [30]. In a high-resolution image, lattice spacing of 0.87 nm was observed, which corresponded to (100) of pyromorphite (Figure 6D). The X-ray diffraction (XRD) pattern from the freeze-dried and powdered cells is shown in Figure 7. The major peaks were explained by those of pyromorphite and hydrocerussite (Pb_3_ (CO_3_)_2_ (OH)_2_) calculated by the crystallographic parameters reported by Mills et al. [30] and Martinetto et al. [31], respectively. 

### 2.4. Evaluation of Ultra-Thin Cross-Sections of KKY-29

To clarify whether the nanoparticles were synthesized inside or outside the cell, ultra-thin cross-sections of KKY-29 cells were prepared using an ultramicrotome and observed by TEM (Figure 8). Cell membranes and organelles were observed under Pb(−) conditions, and the contrast of the images was obscured. In contrast, many nanoparticles were observed inside the KKY-29 Pb(+) cells; thus, nanoparticles were synthesized inside the cells.

### 2.5. Confirmation of Cell Viability

To investigate whether KKY-29 cells survived nanoparticle synthesis, the life and death of the bacterial cells were observed after staining with fluorescent reagents. After treatment with the reagents, green fluorescence was observed when the cells were alive, while red fluorescence was observed when the cells were dead. Differential interference images, green fluorescence images (emission wavelength 520 nm), and red fluorescence images (emission wavelength 604 nm) of KKY-29 before and after treatment with lead acetate are shown in Figure 9. The results were similar before and after the treatment with lead acetate. Rod-shaped cells were observed in the differential interference images (Figure 9A(i),B(i)). Many cells were observed in the green fluorescence images (Figure 9A(ii),B(ii)), while no cells were observed in the red fluorescence images (Figure 9A(iii),B(iii)). These results indicate that KKY-29 could survive nanoparticle synthesis.

### 2.6. Evaluation of the Effect of Phosphate Concentration

Nanoparticles synthesized by KKY-29, cultured in various concentrations of phosphorous, were observed by TEM and analyzed by XRD. After culturing in synthetic medium with 0, 1.0, 10, and 100 mg/mL of phosphorous (conditions A to D), KKY-29 cells were washed with sterile water and incubated with lead acetate solution. After incubation, many nanoparticles were observed in the cells (Figure 10). In the TEM image of condition D, fewer nanoparticles were observed than those in the other samples. The XRD spectra of KKY-29 cells after freeze-drying and crushing with a mortar and pestle are shown in Figure 11. The results revealed that the nanoparticles synthesized under condition A were pyromorphite, while the nanoparticles of condition D were hydrocerussite; the spectra under conditions B and C showed low intensity, and condition B contained both pyromorphite and hydrocerussite.

To investigate the expression of genes related to phosphate metabolism, quantitative PCR was performed. To obtain the cDNA of KKY-29, total RNA extracted from KKY-29 cells was used for reverse transcription. KKY-29 was incubated in lead acetate solution after washing with sterile water and cultured in media containing various concentrations of lead acetate. We analyzed ppK, ppX, and phoD expression (Appendix A). The expression of each gene was standardized with the housekeeping gene *girB*. qPCR analysis using the ΔΔC_T_ method showed that each gene was highly expressed in KKY-29 in 50 µM lead acetate medium or 50 µM lead acetate solution. In contrast, the expression levels were significantly decreased in 5000 µM lead acetate solution (Figure 12).

### 2.7. Various Metal Nanoparticles Synthesized by KKY-29

As KKY-29 demonstrates metal tolerance and synthesizes Pb nanoparticles by decomposing organic phosphoric acid through phosphoric acid transportation, we anticipated a capacity for synthesis of metals other than lead. KKY-29 was suspended in (CH_3_COO)_2_Ca, FeCl_3_, NiCl_2_, CuSO_4_, ZnSO_4_, Na_2_PdCl_4_, AgNO_3_, CdCl_2_, K_2_PtCl_4_, and KAuCl_4_ solutions at final concentrations of 0.05, 0.5, and 5.0 mM. The results are shown in Appendix A. No color change was observed in any of the samples. The samples containing 1.0 mM Na_2_PdCl_4_, K_2_PtCl_4_, and KAuCl_4_ showed some coloration attributed to the metal salt solution. A red coloration observed in the 1.0 mM AgNO_3_ sample and the purple coloration observed in the 0.5 mM KAuCl_4_ sample were consistent with the colors derived by surface plasmon resonance of Ag or Au nanoparticles.

Each sample was observed using TEM (Appendix A). Nanoparticles were observed in some samples (Table 1). The nanoparticles were synthesized inside the cells in most of the samples, while some nanoparticles were synthesized outside the cells in the samples containing 5.0 mM of CdCl_2_ and KAuCl_4_.

The concentration of each metal in the supernatant was measured using ICP-MS. As a negative control, the concentration of the metal salt solution without KKY-29 was measured. The relative values are shown in Appendix A, where the control is set to 1. Approximately 80% of the metal ions were recovered by KKY-29 under all conditions. In particular, KKY-29 recovered a considerable proportion of Au, and 93.8% of Au was recovered from the 5.0 mM solution.

## 3. Discussion

In this study, we used bouillon medium with beef extract as a carbon source for screening. A culture medium with a simple composition and inexpensive preparation would facilitate its application in industrial metal recovery. Some lead nanoparticles such as PbS, PbS_2_, and CsPbBr_3_ have fluorescence characteristics [32,33,34]. However, microorganisms may release substances with fluorescent properties. To confirm the synthesis of CdSe quantum dots with fluorescent properties, it is necessary that the fluorescent properties of bacterial suspensions be measured as a control to verify the absence of endogenous fluorescent substances [35]. We compared the fluorescence intensity of bacterial suspensions with and without lead to screen nanoparticle synthesizing strains. Phylogenetic analysis showed that KKY-29 was closely related to *P*. *koreensis* JCM 14769, originally isolated from farmland soil in Korea. *P*. *koreensis* CPSB21 has chromium tolerance and promotes chromium uptake in plants [36]. In addition, *P. koreensis* AGB-1 shows tolerance to various metals, such as zinc and lead, and accumulates metals [37]. AGB-1 also accumulates lead ions, but the lead ions are fixed as metal complexes outside the cell. The differences in the presence of lead ions suggest that KKY-29 concentrates lead ions by a mechanism different from that of AGB-1.

The LIVE/DEAD Biofilm Viability Kit was used to evaluate the viability of KKY-29 cells. As this kit consists of SYTO9, a membrane-permeable green fluorescent nucleic acid staining reagent, and propidium iodide, a membrane-impermeable red fluorescent nucleic acid staining reagent, the viability of bacteria can be evaluated by two-color fluorescence. In this study, KKY-29 survived after suspension in lead acetate solution. Lead is toxic not only to humans but also to bacteria, in which certain metabolic pathways are selectively inhibited. Consequently, local microbial diversity is reduced as non-resistant microorganisms are eliminated by the release of lead into the environment [38,39].

Pyromorphite is a mineral that is also called apatite [40]. Apatite compounds are valuable catalysts and can be used for the immobilization of toxic metals [41,42]. Among them, pyromorphite is considered to be the most stable mineral species for accumulating lead in the environment because of its insolubility and non-bioavailability. It is widely studied as a form of final precipitate in immobilizing lead in the environment [43,44,45]. However, the method of adding phosphate to a polluted environment shows low conversion efficiency to pyromorphite. For example, in the soil of a shooting range (a typical example of a lead-contaminated environment) about 30% of the phosphate becomes green lead ore, while the rest of the lead remains in a different chemical form [46]. Moreover, phosphate binds to aluminum and iron ions and precipitates [47]. By changing the culture conditions, the KKY-29 strain can recover a large proportion of lead from the environment and deposit it as pyromorphite, and, given further research and development, could be used in the bioremediation of lead-contaminated environments. As organic phosphate is abundant in soil, the use of inorganic phosphorus obtained by decomposition for lead deposition is being studied. *Aspergillus niger* precipitates lead as pyromorphite by the secretion of phytase [48]. 

As KKY-29 synthesizes pyromorphite, consisting of lead and phosphate, KKY-29 is considered to metabolize phosphate and use it for the synthesis of pyromorphite nanoparticles. Therefore, to investigate the effect of phosphoric acid conditions on the synthesis of nanoparticles, a synthesis medium was used. When cultured under phosphorus-free conditions, hydrocerussite nanoparticles were synthesized; pyromorphite nanoparticles were synthesized when cultured under phosphorus-rich conditions. When phosphorus was present in the medium, KKY-29 reproduced using the phosphorus carried over from the preculture medium and phosphorus in the medium, and pyromorphite nanoparticles were synthesized using excess phosphorus. When KKY-29 was cultured under phosphorus-free conditions, the phosphorus carried over from the preculture medium was used only for reproduction and was not hydrolyzed to inorganic phosphate, and the hydrocerussite nanoparticles were synthesized by combining lead ions with carbonate ions generated from the respiratory chain. The KKY-29 strain is thought to have acquired tolerance to lead ions by sequestering them through crystallization to prevent them from causing toxic damage.

## 4. Materials and Methods

### 4.1. Collection and Culture of Microorganisms

We collected samples of fresh water from around the Kamioka mine in Hida city, Gifu prefecture and Okoya mine in Komatsu city, Ishikawa prefecture in Japan (Figure 13). The water samples were scooped up from the surface of water using 50 mL sterile tubes at 14 spots. Collected samples were stocked at 4 °C. In order to isolate the microorganisms with lead tolerance, the water was spread on nutrient glucose agar with lead acetate (1% meat extract/1% peptone/0.1% NaCl/1% glucose/1 mM lead acetate/1.5% agar). The mediums were kept at 25 °C and cultured for several days. The single colonies were transplanted onto another agar medium without lead acetate using sterilized toothpicks and incubated at 25 °C. A total of 96 microorganisms were isolated and named as KKY-1 to KKY-96. 

### 4.2. Screening

For screening of strains with lead nanoparticle synthesis abilities, isolated microorganisms were cultured in nutrient glucose liquid medium with 5.0 mM lead acetate at 25 °C for 24 h. 

As some lead nanoparticles show fluorescence, fluorescence intensity (excitation/emission wavelength 400/460 nm) of each strain was measured by fluorometer (FP-6500, JASCO, Tokyo, Japan). Some strains with strong fluorescence intensity were observed by TEM (JEM-1010, JEOL, Tokyo, Japan) operated at 120 kV to observe the cells and nanoparticle synthesis. For TEM observation, 2 µL of the suspensions of microorganisms were dropped on a grid coated with carbon powder and dried.

### 4.3. Identification of the Strain

In order to identify the strain, the DNA was extracted by the alkaline heat extraction method. At first, a small number of colonies in the agar plate were collected by sterile micropipette tip and suspended in 50 µL of sterile water. The suspension was mixed with 50 µL of 100 mM NaOH and heated at 100 °C for 10 min. After heat treatment, the solution was mixed with 11 µL of 1 M Tris-HCl (pH 7.0) and centrifuged at 14,000 rpm for 1 min. Finally, the supernatant of the mixture was used as the DNA extraction solution. To amplify the 16S rRNA gene, the DNA extraction solution was provided for PCR. The PCR assay was performed in a final volume of 10 μL containing 0.05 μL TaKaRa Ex Taq, 1 μL 10× Ex Taq Buffer, 0.8 μL dNTP mixture, 1.0 μL each primer, 1 μL DNA sample, and 5.15 μL deionized water. The used primers were bacterial 16S rRNA gene primers 10F-5′GTTTGATCCTGGCTCA3′ and 1500R-5′TACCTTGTTACGACTT3′. The reaction mixture was subjected to 35 cycles of amplification as follows: denaturation at 96 °C for 30 s, annealing at 58 °C for 30 s, and extension at 72 °C for 1 min 30 s. After the PCR assay, the solution was subjected to ethanol precipitation for purification. The sequence of purified PCR products was analyzed by FASMAC Co., Ltd. (Kanagawa, Japan). The 16S rRNA gene-related species were obtained using the “Sequence Match” based on type species at NCBI in the web tool RDP. The genealogical trees constructed by the neighbor-joining method were produced by MEGA (Molecular Evolutionary Genetics Analysis).

### 4.4. Synthesis of Nanoparticles

To calculate the correlation between OD_600_ and dry bacterial cell weight, the bacterial cells were centrifuged and washed by sterile water and dried at 150 °C overnight. When screening microorganisms with nanoparticle synthesis ability, nanoparticles were synthesized in the culture medium. In order to remove the medium, the bacterial cells were centrifuged and washed with sterile water. The precipitate of bacterial cells was suspended in sterile water and mixed with various concentrations of lead acetate solution. The mixture solution was incubated on a shaker at 25 °C for 24 h.

### 4.5. Observation of Nanoparticles

To clarify the synthesized nanoparticles in detail, they were analyzed by TEM with EDS andXRD measurements. 

TEM analysis was conducted using a JEM-2010 (JEOL, Tokyo, Japan) operated at 200 kV. A 2 µL solution was dropped on holey carbon film supported by a copper grid. After drying, the grid was examined by TEM. The particle sizes observed by TEM were measured by ImageJ software [49]. Acquired SAED patterns from the nanoparticles were compared with calculated patterns using a self-made program [50] to identify the mineral phase. The freeze-dried bacterial cells were crushed with a pestle/mortar and placed on a nonreflecting silicon holder for XRD. XRD patterns were collected using a RINT-Ultima+ diffractometer (RIGAKU, Tokyo, Japan) with a 1D high-speed X-ray detector (Rigaku D/teX Ultra2) and X-ray tube generating CuKα radiation at 40 kV and 30 mA. CuKβ was attenuated by a Ni foil of 15 µm thickness.

### 4.6. Confirmation of Survival State of Bacterial Cells

To investigate whether KKY-29 was still alive after nanoparticle synthesis, the life and death of the bacterial cells were observed using FilmTracer^TM^ LIVE/DEAD Biofilm Viability Kit (Thermo Fisher Scientific, Waltham, MA, USA). The bacterial cells incubated with and without lead acetate solution were stained by the kit according to the manufacturer’s protocol. Stained bacterial cells were observed by a confocal laser microscope (FV1200 IX83, Olympus, Tokyo, Japan). We took three images for each sample: a differential interference image, green fluorescence image (excitation/emission wavelength 482/520 nm), and red fluorescence image (excitation/emission wavelength 490/604 nm). 

### 4.7. Preparation of Ultra-Thin Cross-Sections of Bacterial Cells

The bacterial cells were suspended with 1-hexadecene and frozen by high pressure freezer (HPM-010, Balzers, Liechtenstein, BALTEC). Frozen cells were fixed in 2% osmium tetroxide acetonitrile solution at −80 °C for 3 days. After washing in anhydrous acetone, the cells were embedded in spurr resin. Ultra-thin sections were prepared by using an ultramicrotome fitted with a glass knife. The samples were stained with 4% uranyl acetate and 4% lead citrate before TEM observation. TEM analysis was conducted using a JEM-1010 operated at 120 kV.

### 4.8. Measurement of Lead Concentration by ICP-MS

After nanoparticle synthesis, lead concentration of the supernatant was measured by ICP-MS. Various concentrations of KKY-29 (final concentrations: 0, 100, 500, 1000 mg/mL) were incubated in various concentrations of lead acetate solution (final concentrations: 0.05, 0.5, 5.0 mM). The bacterial suspension in lead acetate solution was centrifuged (4 °C, 4000 g, 10 min). The supernatant was filtered by membrane filters of 0.45 µm pore size and decomposed with nitric acid. One milliliter of the filtrate was mixed with concentrated nitric acid and heated at 120 °C until evaporation twice. The decomposed products were dissolved and filled to 10 mL by 0.08 M nitric acid solution. The samples were subjected to ICP-MS analysis.

### 4.9. Synthesis of Nanoparticles Other than Lead

Bacterial cells were washed with sterile water and suspended in various metal solutions ((CH_3_COO)_2_Ca, FeCl_3_, NiCl_2_, CuSO_4_, ZnSO_4_, Na_2_PdCl_4_, AgNO_3_, CdCl_2_, K_2_PtCl_4_ and KAuCl_4_) with final concentrations of 50, 500 and 5000 µM. All metal salt solutions were sterilized by membrane filters of a 0.45 µm pore size. The bacterial cells after incubation were observed by TEM and subjected to XRD measurement. Furthermore, the metal concentration of supernatant was measured by ICP-MS.

### 4.10. Total DNA Isolation and Genome Sequencing

For genome sequencing, the plasmids of the bacteria were extracted and purified using the SDS-alkaline denaturation method. In order to analyze the genome sequence, the purified plasmids were dissolved in tris-EDTA (pH 8.0) and sent to BGI JAPAN (Hyogo, Japan).

### 4.11. Quantitative PCR

KKY-29 incubated in nutrient glucose medium with lead acetate at 25 °C for 24 h and KKY-29 incubated in lead acetate solution at 25 °C for 24 h were put into 1.5 mL tubes. Total RNA of the samples was extracted using Sepasol RNA I Super G (Nacalai Tesque Inc., Kyoto, Japan) according to the manufacturer’s protocol and dissolved with TE buffer. cDNA was synthesized by reverse transcription of total RNA using PrimeScript RT reagent Kit (TaKaRa, Tokyo, Japan) according to the manufacturer’s protocol. Gene-specific primers for qPCR were designed by Primer BLAST (https://www.ncbi.nlm.nih.gov/tools/primer-blast/ accessed on 2 November 2021) with default settings except for PCR product size (150–200 bp). qPCR reaction was performed using Thunderbird Probe qPCR Mix (TaKaRa, Tokyo, Japan). The results of qPCR analysis were obtained using the Comparative C_T_ method (ΔΔC_T_ method). The used primers are shown in Table 2.

## 5. Conclusions

In this study, the strain KKY-29 was identified as a bacterium with lead resistance and the ability to synthesize nanoparticles from fresh water collected from an abandoned mine. KKY-29 belongs to the *Pseudomonas* genus and synthesizes pyromorphite and hydrocerussite nanoparticles from a lead acetate solution. As KKY-29 survived in lead acetate, it could be applied in fresh-water bioremediation strategies. 

Figure 14 presents a schematic diagram of nanoparticle synthesis by KKY-29. KKY-29 absorbed and crystallized lead ions as nanoparticles using phosphate ions generated from the metabolization of phosphate and carbonate ions.

## Figures and Tables

**Figure 1 ijms-23-02483-f001:**
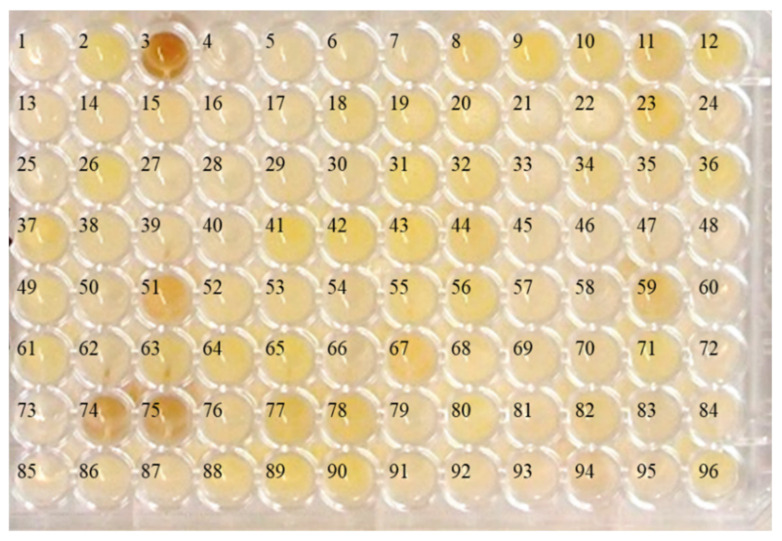
The color result of the screening. The microorganisms from KKY-1 to KKY-96 were incubated in nutrient medium with lead acetate.

**Figure 2 ijms-23-02483-f002:**
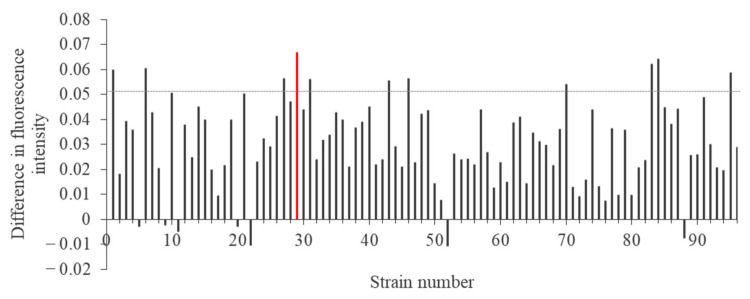
Difference in fluorescence intensity between KKY-1 and KKY-96 strains cultured in lead acetate (excitation: 400 nm, fluorescence: 460 nm). The strains with a difference in fluorescence intensity greater than about 0.05, indicated by the gray dotted line, were subjected to further observations.

**Figure 3 ijms-23-02483-f003:**
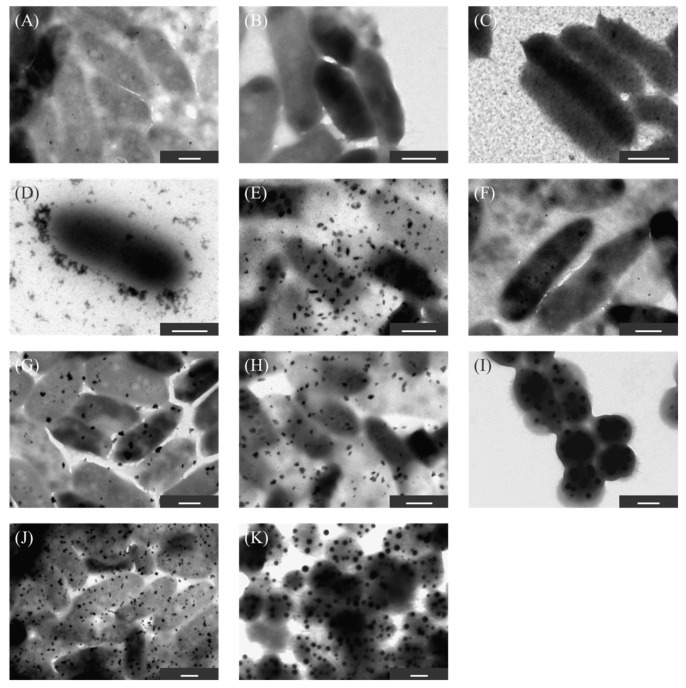
TEM images of the 11 strains incubated in the medium with lead acetate. (**A**) KKY-1, (**B**) KKY-6, (**C**) KKY-27, (**D**) KKY-29, (**E**) KKY-31, (**F**) KKY-43, (**G**) KKY-46, (**H**) KKY-70, (**I**) KKY-83, (**J**) KKY-84, (**K**) KKY-95. Scale bar is 500 nm.

**Figure 4 ijms-23-02483-f004:**
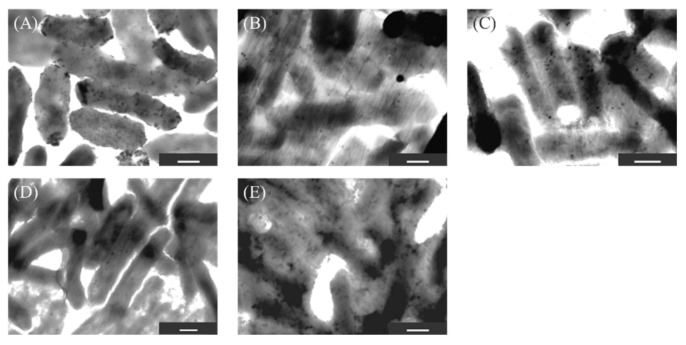
TEM images of the five strains suspended in the lead acetate solution. (**A**) KKY-29, (**B**) KKY-31, (**C**) KKY-46, (**D**) KKY-70, (**E**) KKY-84. Scale bar is 500 nm.

**Figure 5 ijms-23-02483-f005:**
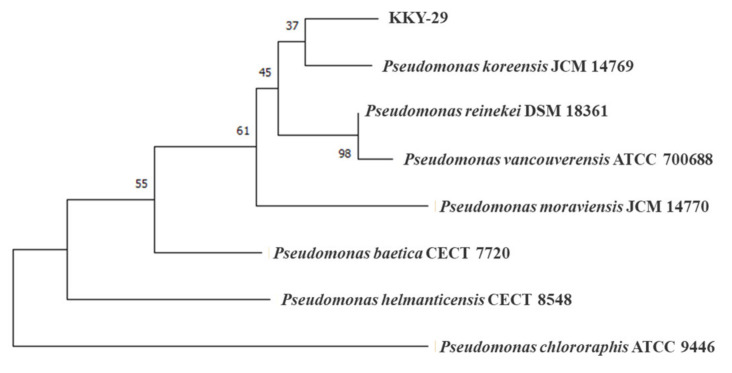
The phylogenetic tree analysis of KKY-29.

**Figure 6 ijms-23-02483-f006:**
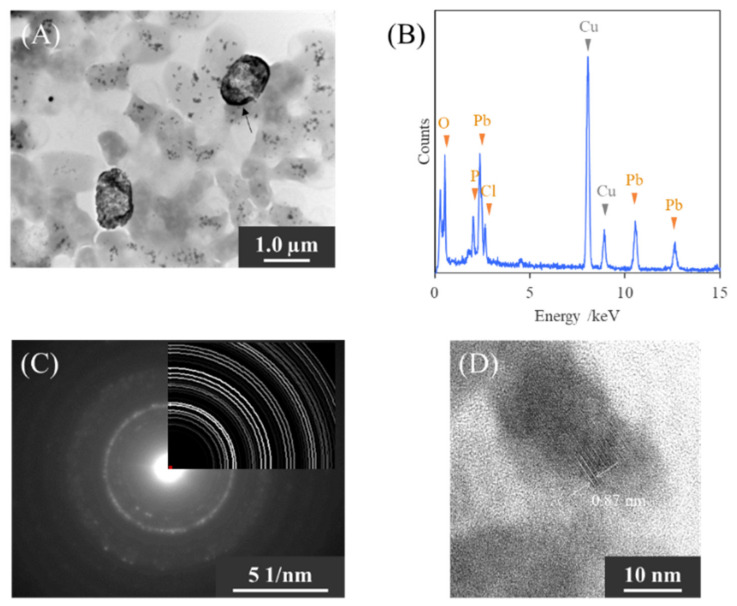
(**A**) TEM image of KKY-29 (Pb(+) condition). (**B**) X-ray spectrum from the nanoparticles indicated in (**A**) with an arrow. X-ray of Cu was originated from the supporting grid. (**C**) SAED pattern from the cell indicated in (**A**) with an arrow. The inset at the top-right is the calculated ring pattern for pyromorphite. (**D**) Lattice image of the nanoparticles indicated in (**A**) with an arrow. The lattice spacing of 0.87 nm corresponds to (100) of pyromorphite.

**Figure 7 ijms-23-02483-f007:**
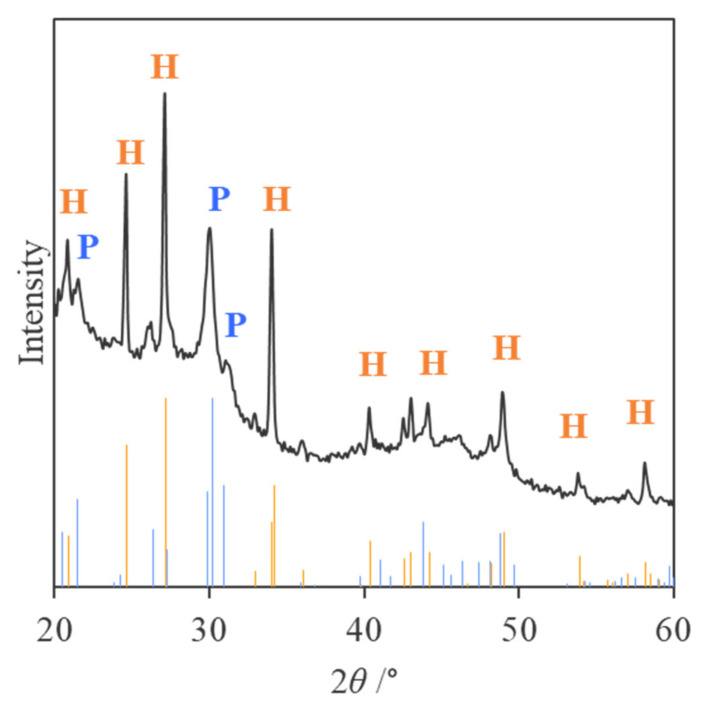
XRD pattern of nanoparticles synthesized by KKY-29. The blue and orange bars at the bottom show the peak positions and intensities expected for pyromorphite and hydrocerussite, respectively.

**Figure 8 ijms-23-02483-f008:**
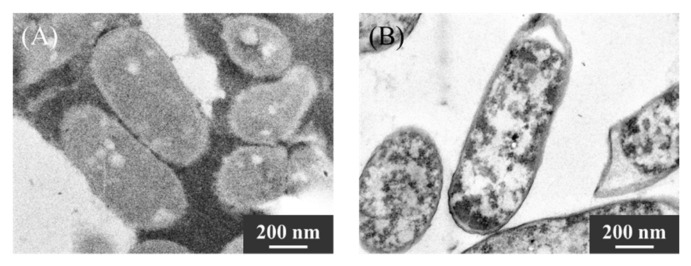
TEM images of ultra-thin cross-sections of bacterial cells (**A**) Pb(−) condition, (**B**) Pb(+) condition.

**Figure 9 ijms-23-02483-f009:**
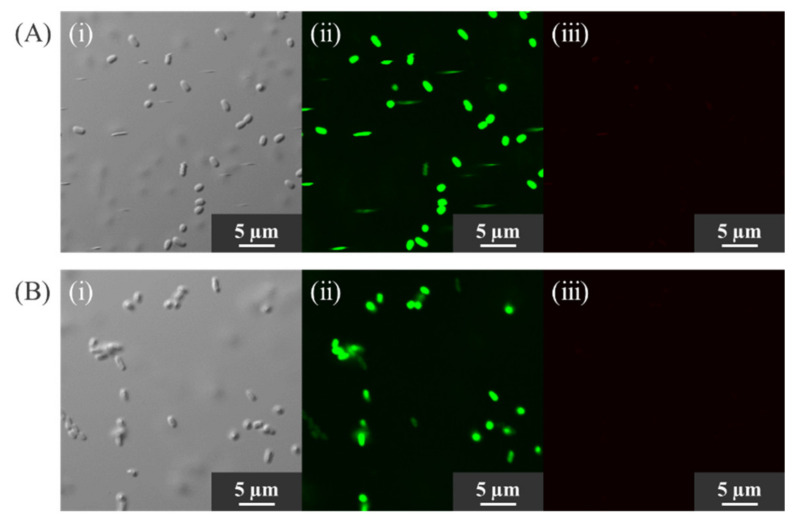
Optical microscope images of bacterial cells. (**A**) Pb(−) condition, (**B**) Pb(+) condition. (**i**) Differential interference images. (**ii**) Fluorescent images (Em. 520 nm). (**iii**) Fluorescent images (Em. 604 nm).

**Figure 10 ijms-23-02483-f010:**
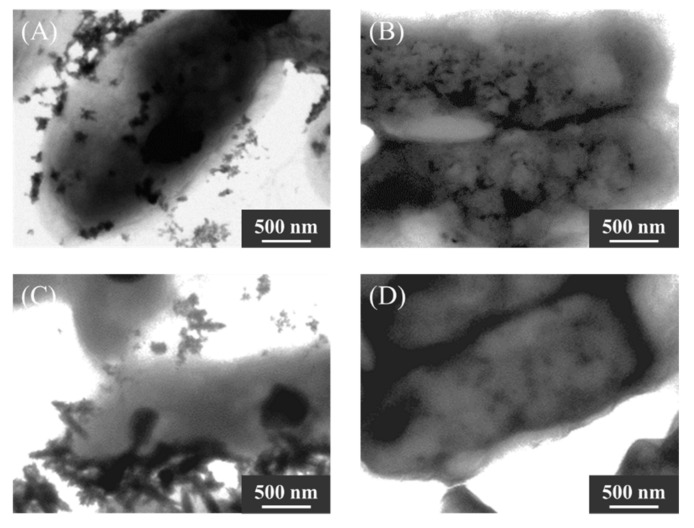
TEM images of nanoparticles synthesized by KKY-29 incubated in various concentrations of phosphorous ((**A**) 0 mg/mL, (**B**) 1.0 mg/mL, (**C**) 10 mg/mL, (**D**) 100 mg/mL).

**Figure 11 ijms-23-02483-f011:**
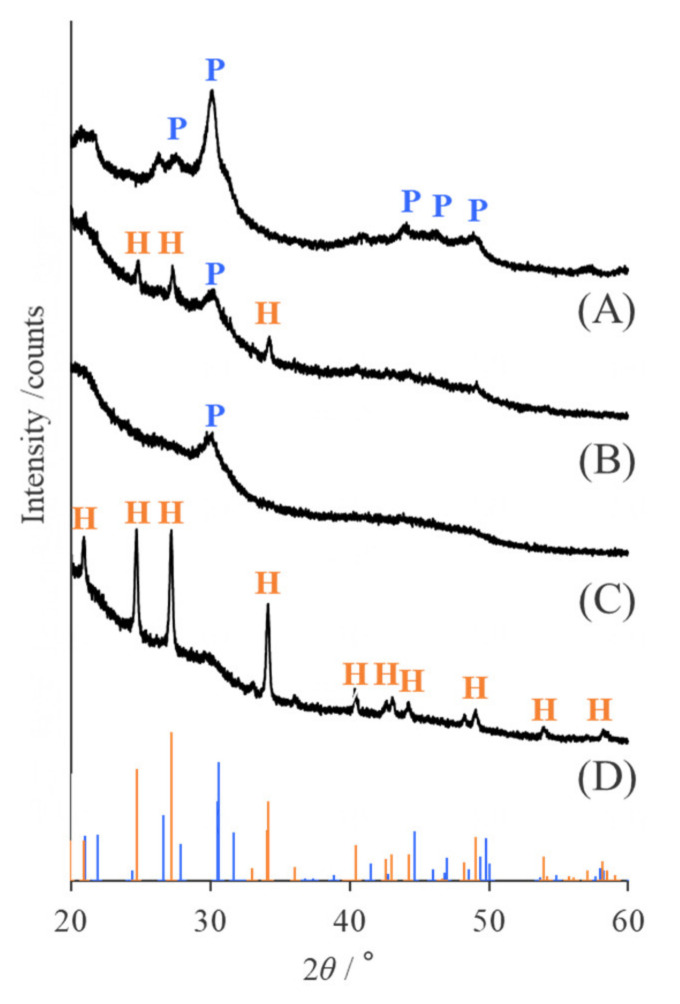
XRD patterns of the nanoparticles synthesized by KKY-29 incubated in various concentrations of phosphorous ((**A**) 100 mg/mL, (**B**) 10 mg/mL, (**C**) 1.0 mg/mL, (**D**) 0 mg/mL). The black line represents the pattern of each sample, and blue and orange bars at the bottom show the peak positions and intensities expected for pyromorphite and hydrocerussite, respectively.

**Figure 12 ijms-23-02483-f012:**
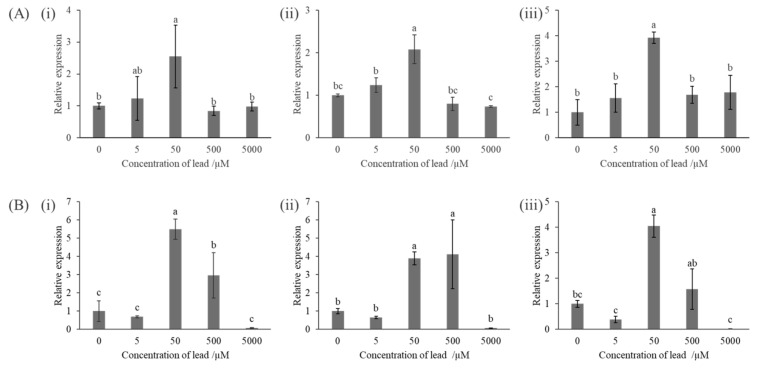
Expression analysis of genes related to phosphate metabolism. ((**i**) *ppk*, (**ii**) *ppx*, (**iii**) *phoD*) (**A**) KKY-29 incubated in various concentrations of lead medium. (**B**) KKY-29 shaken in various concentrations of lead solution.

**Figure 13 ijms-23-02483-f013:**
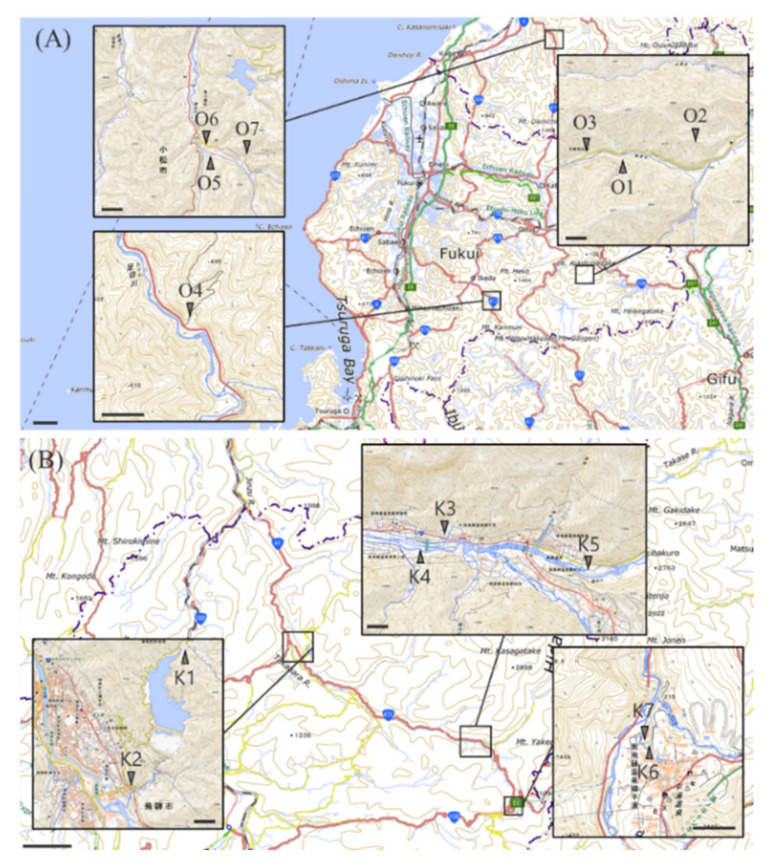
The spots where the fresh water was collected: (**A**) around the closed Okoya mine, (**B**) around the closed Kamioka mine. The map data are quoted from GSI Maps.

**Figure 14 ijms-23-02483-f014:**
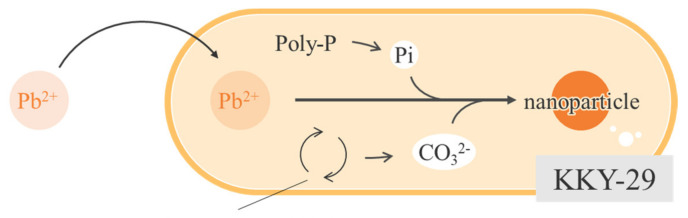
Schematic model of nanoparticles synthesis in KKY-29.

**Table 1 ijms-23-02483-t001:** Summary of synthesis of various metal nanoparticles by KKY-29.

The Concentration of Metal Salt/mM	Ca	Fe	Ni	Cu	Zn	Pd	Ag	Cd	Pt	Au
5.0	65.5 ± 10.9	-	-	-	-	-	24.8 ± 5.66	-	-	-
0.5	81.2 ± 15.5	-	41.9 ± 8.88	-	-	-	7.93 ± 6.74	40.1 ± 5.81	-	27.8 ± 6.45
0.05	74.8 ± 14.8	65.0 ± 14.0	-	-	-	>100	-	-	56.5 ± 14.3	-

The number represents the average diameter of synthesized nanoparticles (nm) and the hyphen represents no nanoparticle synthesis.

**Table 2 ijms-23-02483-t002:** Sequences of specific primers used in qPCR.

Primer Name	Sequences
ppX_F	AGTGAATGCCGAAGGTCTGG
ppX_R	CAGTGATCCATGCGTTGCAG
ppk_F	TGGTGCACGGTCATGTAGAC
ppk_R	CGATCTCGTCGCGGAAGTAA
phoD_F	GTGCGCAATTTCGTGTTCCT
phoD_R	AGGTCTTGTCCAGCGGATTG
gyrB_F	TCCGAACTGTACCTCGTGGA
gyrB_R	GCGCAACTTGTCGATGTTGT

## Data Availability

Not applicable.

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
