# Peer review of "Deposition of Lead Phosphate by Lead-Tolerant Bacteria Isolated from Fresh Water near an Abandoned Mine"

_ijms, 2022, doi:10.3390/ijms23052483_

Round 1

Reviewer 1 Report

There are some problems in this manuscript. Revision is needed to make this manuscript suitable for publication.

  1. IIn 77-78,“The color result of the screening. The microorganisms from KKY-1 to KKY-92 were incubated in nutrient medium with lead acetate”,Is "KKY-1 to KKY-92" spelled incorrectly?
  2. Fig 4 does not demonstrate that bacteria can synthesize nanoparticles in lead acetate solution.
  3. There is no Fig 5 in the article.
  4. A significance analysis is suggested for the Expression of gene related to phosphate metabolism in Fig. 12.

Author Response

1. IIn 77-78,“The color result of the screening. The microorganisms from KKY-1 to KKY-92 were incubated in nutrient medium with lead acetate”,Is "KKY-1 to KKY-92" spelled incorrectly?

(response)

Thank you for pointing out our mistakes. We corrected "KKY-1 to KKY-92" to "KKY-1 to KKY-96".

2. Fig 4 does not demonstrate that bacteria can synthesize nanoparticles in lead acetate solution.

(response)

I am sorry that the legend of Figure 4 was not enough to understand the TEM images. In Figure 4A~E, to screen the strains which synthesize nanoparticles, 5 strains were incubated in lead acetate solution without medium. As black particles were observed in TEM observation, which indicated that KKY-29, 46, and 84 strains synthesized nanoparticles. Detailed analysis of KKY-29 showed that the lead nanoparticles were synthesized in lead acetate solution.

3. There is no Fig 5 in the article.

(response)

Thank you for pointing out our mistakes. We corrected “Figure 4. The phylogenetic tree analysis of KKY-29” to “Figure 5”.

4. A significance analysis is suggested for the Expression of gene related to phosphate metabolism in Fig. 12.

(response)

Thank you for your comment.

Reviewer 2 Report

The manuscript that has been reviewed discusses very reliably the isolation of bacteria and subsequent use of this microorganism for the deposition of lead phosphate and bio-formation of nanoparticles. This work was written carefully. The results were presented logically. The low number of cited publications from 2020-2021 is noteworthy. This needs to be completed.  

Author Response

The manuscript that has been reviewed discusses very reliably the isolation of bacteria and subsequent use of this microorganism for the deposition of lead phosphate and bio-formation of nanoparticles. This work was written carefully. The results were presented logically. The low number of cited publications from 2020-2021 is noteworthy. This needs to be completed.

(response)

Thank you for your advice. We added references 4, 21 and 27. In addition, we cited literature on nanoparticle synthesis by metal-tolerant bacteria and bioremediation by nanoparticle-synthesizing bacteria and referred to references 22-26.

Round 2

Reviewer 1 Report

 Accept in present form